# A Large Gap in Patients’ Characteristics and Outcomes between the Real-World and Clinical Trial Settings in Community-Acquired Pneumonia and Healthcare-Associated Pneumonia

**DOI:** 10.3390/jcm11020297

**Published:** 2022-01-07

**Authors:** Nobuhiro Asai, Yuichi Shibata, Daisuke Sakanashi, Hideo Kato, Mao Hagihara, Yuka Yamagishi, Hiroyuki Suematsu, Hiroshige Mikamo

**Affiliations:** 1Department of Clinical Infectious Diseases, Aichi Medical University Hospital, Nagakute 480-1195, Aichi, Japan; nobuhiro0204@gmail.com (N.A.); shibata.yuuichi.414@mail.aichi-med-u.ac.jp (Y.S.); saka74d@aichi-med-u.ac.jp (D.S.); katou.hideo.233@mail.aichi-med-u.ac.jp (H.K.); hagimao@aichi-med-u.ac.jp (M.H.); y.yamagishi@mac.com (Y.Y.); hsuemat@aichi-med-u.ac.jp (H.S.); 2Department of Pathology, University of Michigan, Ann Arbor, MI 48109, USA; 3Department of Pharmacy, Mie University Hospital, Tsu 514-0001, Mie, Japan; 4Department of Clinical Pharmaceutics, Division of Clinical Medical Science, Mie University Graduate School of Medicine, Tsu 514-0001, Mie, Japan; 5Department of Molecular Epidemiology and Biomedical Sciences, Aichi Medical University, Nagakute 480-1195, Aichi, Japan

**Keywords:** pneumonia, antibiotics, clinical trial, evidence-based medicine, real world

## Abstract

(1) Introduction: Evidence-based medicine (EBM) is necessary to standardize treatments for infections because EBM has been established based on the results of clinical trials. Since entry criteria for clinical trials are very strict, it may cause skepticism or questions on whether the results of clinical trials reflect the real world of medical practice. (2) Methods: To examine how many patients could join any randomized clinical trials for the treatment of community-acquired pneumonia (CAP) and healthcare-associated pneumonia (HCAP). We reviewed all the pneumonia patients in our institute during 2014–2017. The patients were divided into two groups: patients who were eligible for clinical trials (participation-possible group), and those who were not (participation-impossible group). Exclusion criteria for clinical trials were set based on previous clinical trials. (3) Results: A total of 406 patients were enrolled in the present study. Fifty-seven (14%) patients were categorized into the participation-possible group, while 86% of patients belonged to the participation-impossible group. Patients in the participation-possible group had less comorbidities and more favorable outcomes than those with the participation-impossible group. As for the outcomes, there were significant differences in the 30-day and in-hospital mortality rates between the two groups. In addition, the participation-possible group showed a longer overall survival time than the participation-impossible group (*p* < 0.001 by *Log-Rank* test). (4) Conclusion: There is a difference in patients’ profile and outcomes between clinical trials and the real world. Though EBM is essential to advance medicine, we should acknowledge the facts and the limits of the clinical trials.

## 1. Introduction

Evidence-based medicine (EBM) aims to assist physicians in making rational decisions in general practices. As EBM is established according to the results of clinical trials, clinical trials are considered one of the essential undertakings and are put at the top of priority among physicians in constructing therapeutic strategies [1]. A randomized control trial (RCT) evaluates the efficacy and tolerability of a new antibiotic treatment, and EBM guidelines/recommendations are made based on those results. There is no room for doubt that current medicine is based on EBM. However, we skeptically think about that when we consider eligibility of pneumonia patients for EBM guidelines or recommendations in actual practice. Entry criteria for any clinical trial are generally very strict, and most patients might not be suitable for the studies. Thus, it is reasonable to doubt whether the results of clinical trials reflect the real world in general practice. We already reported that only 24% of candidemia patients could be eligible in a clinical trial [2]. Pneumonia remains a leading cause of infection deaths worldwide [3,4]. Particularly, elderly patients with pneumonia tend to have more comorbidities than young patients, and the mortality rate is higher than other groups [4,5]. Since it was found that contact with the healthcare facility is not a strong predictor of risk for multidrug-resistant bacteria, healthcare-associated Pneumonia (HCAP) has been removed from hospital-acquired pneumonia (HAP)/ventilator-associated pneumonia guidelines. However, HCAP in Japan was included in HAP due to the greater patients’ profile diversity of HCAP than CAP [6,7]. We have suspected that there might be a distinct difference of clinical pictures (characteristics) between the patients eligible and those excluded from the study, and thus we decided to perform this study. This study focused on to what degree community-acquired pneumonia (CAP) and HCAP patients are eligible for clinical trials, to investigate whether antibiotic therapy is effective and/or tolerable for these patients. This is the first report demonstrating to what degree clinical data, on which EBM is based on, reflects real-world patients with pneumonia.

## 2. Methods

### 2.1. Study Design

Our institute is a 900-bed tertiary care center and is located in the countryside at Aichi prefecture in central Japan. For the purpose of how many community-onset pneumonia patients in our institute could join any randomized clinical trials for an antibiotic treatment among pneumonia patients, we reviewed all CAP and HCAP patients who were admitted to our hospital between September 2014 and May 2017. Pneumonia was diagnosed according to the previously published international guidelines [8]. CAP and HCAP were categorized based on the criteria published by the American Thoracic Society/Infectious Diseases Society of America (ATS/IDSA) in 2006 [9,10]. Severity of pneumonia was evaluated by A-DROP [10], CURB-65 [11], Pneumonia Severity Index (PSI) [12], I-ROAD [13] and SOFA score [14]. Comorbidity was evaluated by the Charlson comorbidity index (CCI) [15]. The patients were divided into two groups: patients who were eligible for clinical trials (participation-possible group), and those who were not (participation-impossible group). Then, patients’ characteristics (age, sex), pathogens isolated, clinical outcomes such as the treatments, 30-day or in-hospital mortality and the reasons of exclusion from the clinical trial, were evaluated.

### 2.2. Patient Selection

Exclusion criteria commonly used in past ordinary clinical trials are as follows [16,17,18];
(1)Age < 18 years, >80 years;(2)Coexisting comorbidities or medical conditions which are difficult to evaluate for pneumonia such as severe liver dysfunction, severe renal dysfunction or HIV/AIDS (severe liver dysfunction was defined as serum total bilirubin, or aspartate aminotransferase/alanine aminotransferase > the upper limit of the normal reference range × 3. Severe renal dysfunction was defined as creatinine clearance < 30 mL/min). Unassessable pulmonary diseases include viral pneumonia, pneumocystis pneumonia [19,20], mycobacterium infections, eosinophilic pneumonia and interstitial pneumonitis. Unassessable malignancies were defined as any malignancy terminated stage or the one with any metastatic lesion to the lungs and/or receiving palliative therapy. Unassessable diabetes mellitus was defined as serum-hemoglobin A1c (NGSP) ≥ 7.0%;(3)Aspiration pneumonia [21,22];(4)Receiving immunosuppressive therapy due to any cause;(5)Receiving chemotherapy for malignancy;(6)Receiving hemodialysis due to any cause;(7)Poor activities of daily living (ADL) or requiring any help (Eastern Cooperative Oncology Group (ECOG) performance status (PS) ≥ 3) such as needing tube feeding or home oxygen therapy;(8)Having other complicated infection;(9)Requiring mechanical ventilation and/or requiring treatments in the intensive care unit;(10)Poor prognosis (anticipated life expectancy < 90 days or patients who are not expected to survive until the end of the trial);(11)Pregnancy.

This study was approved by the Institutional Review Board of Aichi Medical University Hospital.

### 2.3. Microbiological Evaluation

A sputum sample and two sets of blood were collected from each patient for microbiological examination. Serological tests were performed to detect antibodies against *Mycoplasma pneumoniae* [23] and *Chlamydophila pneumoniae* [24]. Additionally, Legionella pneumophila serogroup 1 antigen in the urine was tested by immunochromatography. The antimicrobial susceptibility of isolated bacterial pathogens was assessed on the basis of the minimum inhibitory concentration according to the Clinical and Laboratory Standards Institute guidelines [25]. Methicillin-resistant *Staphylococcus aureus*, *P. aeruginosa*, *Acinetobacter baumannii*, and extended-spectrum β-lactamase-producing organisms were defined as potentially drug-resistant (PDR) pathogens based on ATS/IDSA guidelines [26].

### 2.4. Definition of Appropriate and Inappropriate Treatment, Initial Treatment Failure

Antibiotic treatment was classified as appropriate or inappropriate according to whether the identified pathogens were sensitive or resistant, respectively, to the initially prescribed antibiotics. Initial treatment failure was defined as death during the initial treatment or a change in the antibiotic regimen from the initial agents within 72 h after starting the treatment due to a lack of response or clinical deterioration (e.g., worsening of fever, respiratory condition or radiologic status; requiring mechanical ventilation, aggressive fluid resuscitation or vasopressors).

### 2.5. Statistical Analyses

The data for categorical variables are expressed as percentages and continuous variables as mean ± standard deviation (SD). Chi-squared or Fisher’s exact test (two-tailed) were used to compare categorical variables, and unpaired Student’s *t* test or Mann–Whitney *U* test to compare continuous variables. Overall survival time (OS) was calculated as from the date of diagnosis until the date of death from any cause. A significance was evaluated by *Log-rank* test. Missing values were evaluated by the missing analysis of the software. Statistical analyses involved use of SPSS version 26 for Windows (SPSS Inc., Chicago, IL, USA). A *p*-value < 0.05 was considered statistically significant.

## 3. Result

A total of 406 patients were enrolled in the present study. Table 1 shows the patients’ characteristics and clinical outcomes. Fifty-seven (14%) patients were categorized into the participation-possible group, while 86% patients were in the participation-impossible group. Comparing the two groups, patients in the participation-possible group have less comorbidities than those with participation-impossible group. The severity of pneumonia was much more severe in patients within the participation-impossible group than in those with the participation-possible group. As for the outcomes, the patients with the participation-possible group had more favorable outcomes than those within the participation-impossible group. Mechanical ventilations and do not attempt resuscitation (DNAR) orders were more frequently seen in participation-impossible group than participation-possible group. PDR pathogens were seen more frequently in the participation-possible group than in those within the participation-impossible group (5% vs. 16%, *p* = 0.032). There were no significant differences in the frequency of antipseudomonal agents use as the initial treatment between the two groups. The duration of antibiotics use was longer in patients within the participation-impossible group than in those in the participation-possible group, while there was no difference of duration of admission between the two groups. As for pathogens isolated, MRSA was more frequently seen in participation-impossible group than in the participation-possible group (20% vs. 0%, *p* = 0.013), while *Haemophillus influenzae* was seen more frequently in the participation-possible group than in the participation-impossible group (35% vs. 8%, *p* = 0.042).

Table 2 shows comparison of patients’ characteristics and outcomes between participation possible and participation-impossible group among CAP patients. Fifty-one patients (29%) were in the participation-possible group. Participation possible groups are older and have more comorbidities than the participation-impossible group. All severity scores of pneumonia were higher in the participation-impossible group than in the participation-possible group. There were no differences in 30-day and in-hospital mortality rates in between the two groups. Mean duration of antibiotic therapy was shorter in the participation-possible group than in the participation-impossible group (12.4 ± 10.5 vs. 17.5 ± 16.0 days, *p* = 0.025), while duration of hospital stay did not differ between the two groups.

Table 3 shows comparison of patients’ characteristics and outcomes between participation pos sible and impossible group among HCAP patients. Only 6 patients (3%) were in the participation-possible group among HCAP patients. Although there were no significant differences in age and all pneumonia severity scores in between the 2 groups, CCI was higher than in the participation-impossible group than in the participation-possible group (0.8 vs. 2.8, *p* = 0.022). There was no differences of 30-day and in-hospital mortality rates in between the two groups. There were no differences in duration of hospital stay or antibiotic treatment between the two groups. As for pathogens isolated, isolation of MRSA and Pseudomonas aeruginosa did not differ between the two groups. On the other hand, H. influenza and Moraxella catarrahis tended to isolate more often in the participation-possible group than in the participation-impossible group.

As for overall survival times, the participation-possible group displayed a longer overall survival times (OSs) than the participation-impossible group (median OS not reached vs. 43.3 months, *p* < 0.001 by *Log-rank* test), as shown in Figure 1.

In the subanalysis of OSs among CAP and HCAP, the participation-possible group among CAP patients showed a longer OSs than the participation-impossible group (median OSs not reached vs. 53.9 months, *p* < 0.001 by *Log-rank* test) (Figure 2), while there were no differences in the participation possible and impossible group among HCAP patients (Figure 3).

In terms of reasons for not being able to join a clinical trial, underlying diseases or conditions which could not be assessed correctly was the most commonly seen in 254 (73%) patients, followed by age in 180 (52%) patients (Table 4).

## 4. Discussion

Patients in the real world are quite different from those who can participate in a clinical trial. We already reported that only 24% of candidemia patients could participate in a clinical trial. Patients who can participate in a clinical trial have better PSs and longer overall survival times than those seen in actual medical practice [2]. In this study, community-onset pneumonia patients within the participation-possible group showed a lesser severity of pneumonia and fewer comorbidities than those in the participation-impossible group. We found that the participation-impossible group had higher 30-day and in-hospital mortality rates than the participation-possible group. In fact, identification of PDR pathogens, mechanical ventilation and Do Not Attempt Resuscitation (DNAR) order were more frequently seen in the participation-impossible group than in the possible group. In Japan, discussing DNAR order with Japanese family members is still considered to be taboo [27]. Therefore, these results could suggest that patients in the participation-impossible group have a worse prognosis than those in the participation-possible group do. It is well-known that HCAP patients are more likely to have worse PSs and more comorbidities than those with CAP [4,5,7]. We should consider a RCT focusing on the elderly or fragile people who are usually excluded from the trials, or analyze alternatives such as propensity-score matching analysis. These will be helpful for clinicians to make a rational decision in treating those patients.

Outstandingly, the OSs in the participation-impossible group with CAP were significantly shorter than those in the participation-possible group, while 30-day and in-hospital mortality rate did not differ between the two groups. More comorbidities could affect the prognosis among the participation-impossible group. Particularly, more aspiration pneumonia was seen in 65/125 (52%) and 140/223 (63%) patients in the participation-impossible group with CAP and HCAP, respectively. Performance status in patients with aspiration pneumonia are likely to decline, and some of them become bedridden [28]. These poor conditions can lead to a lower survival rate in the participation-impossible group. Unfortunately, we did not analyze these data. Physicians should pay attention to them after discharge. Additionally, 97% of HCAP patients in the studies [4,5,7] were excluded from the clinical trial. In addition, HCAP patients in the participation-possible group had much shorter durations of antibiotic treatment and admission than those in the participation-impossible group. An appropriate duration of antibiotics is said to be 5–7 days. A sub-analysis showed that there was no difference in mean duration of antibiotic therapy between the survival and 30-day death groups among HCAP patients (survival 14.2 vs. 30-day death 10.3 days, *p* = 0.21). The results of our study also suggest that HCAP patients are likely to have longer duration of antibiotic therapy, lasting 10–14 days, as we expected. The therapeutic strategy for HCAP patients might have to be reconsidered due to the poor general conditions.

As for an initial antibiotic therapy among CAP patients, more penicillin and less fluoroquinolones were seen in the participation-impossible group than the participation-possible group. The reasons are that the initial antibiotic selections were based on the patients’ characteristics. The patients who received penicillin had aspiration pneumonia in 31/70 (44.3%) of cases, and those who received fluoroquinolones were younger than 50 years in 7/21 (33.3%). The doctors prescribed penicillin and fluoroquinolones to the patients to cover anerobic bacteria and atypical bacteria, respectively.

There are several limitations in our study. First, this is a retrospective study on a small population. Thus, there might be a bias in data selection and analysis, such as the severity of pneumonia, Second, we only evaluated patients who were admitted to our institute. The choice of initial antibiotic therapy, indication of hospitalization, ICU admission and DNAR orders were based on the physicians’ decisions. There might be possibility that patients in this study could not reflect the whole population of pneumonia patients.

## 5. Conclusions

In conclusion, 14% patients could join the clinical trial, while 86% patients could not. There is a difference in patients’ profiles and outcomes between the real world and the clinical trial. Though EBM is very important and essential to advancing medicine, we should acknowledge the facts and limits of clinical trials. Physicians should not be overconfident in EBM based on the results of a clinical trial.

## Figures and Tables

**Figure 1 jcm-11-00297-f001:**
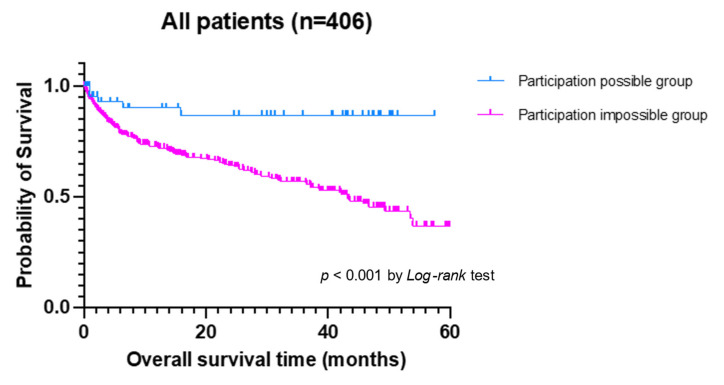
The comparison of overall survival time (OS) among community-onset pneumonia patients according to participation-possible group (blue line) and participation-impossible group (pink line).

**Figure 2 jcm-11-00297-f002:**
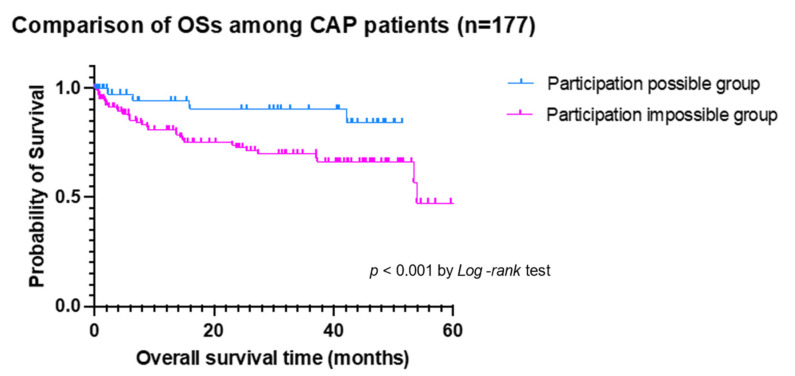
The comparison of overall survival time (OS) among CAP patients according to participation-possible group (blue line) and participation-impossible group (pink line).

**Figure 3 jcm-11-00297-f003:**
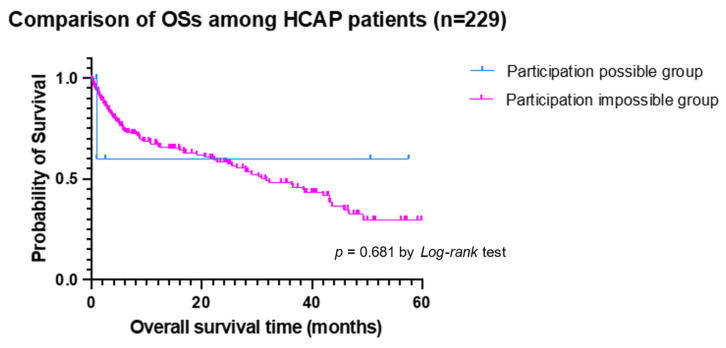
The comparison of overall survival time (OS) among HCAP patients according to participation-possible group (blue line) and participation-impossible group (pink line).

**Table 1 jcm-11-00297-t001:** Comparison of patients’ characteristics and outcomes between the participation-possible group and impossible group.

Variables	All Patients (n = 406)	Participation-Possible Group (n = 57)	Participation-Impossible Group (n = 349)	*p*-Value
Mean age (years ± SD)	75.4 ± 14.8	54.9 ± 17.6	78.8 ± 11.2	<0.001
Median age (years, range)	79 (18–103)	56 (18–79)	81 (37–103)	-
Male gender (n, %)	257 (63)	28 (49)	229 (66)	0.017
Smoking history (n, %)				
Current smoker	36 (9)	11 (19)	25 (7)	0.003
Ex-smoker	205 (50)	24 (42)	181 (52)	0.172
Never smoker	135 (33)	21 (37)	114 (33)	0.535
Unknown	30 (7)	1 (2)	29 (8)	0.079
Underlying diseases (n, %)				
Heart disease	126 (31)	4 (7)	122 (35)	<0.001
Chronic pulmonary disease	175 (43)	20 (35)	155 (44)	0.187
Diabetes mellitus	61 (15)	1 (2)	60 (17)	0.001
Chronic kidney disease	51 (13)	0	51 (15)	0.002
Hemodialysis	16 (4)	0	16 (5)	0.099
Hepatic disease	14 (3)	0	15 (4)	0.111
Collagen vascular disease	41 (10)	0	41 (12)	0.006
Cerebrovascular disease	100 (25)	0	100 (29)	<0.001
Malignancy	74 (18)	0	75 (21)	<0.001
Dementia	74 (18)	2 (4)	72 (21)	0.002
Gastroesophageal reflux disease	14 (3)	3 (5)	11 (3)	0.418
Proton pump inhibitor use	122 (30)	5 (9)	117 (34)	<0.001
Sleep agents use	60 (15)	0	60 (17)	<0.001
Charlson comorbidity index (mean ± SD)	2.1 ± 1.8	0.4 ± 0.5	2.4 ± 1.9	<0.001
Charlson comorbidity index ≥ 3 (n, %)	120 (30)	0	121 (35)	<0.001
Category of pneumonia (n, %)				
Community-acquired pneumonia	177 (44)	51 (89)	126 (36)	<0.001
Healthcare-associated pneumonia	229 (56)	6 (11)	223 (64)	
Severity of pneumonia (mean ± SD)				
A-DROP score	2.0 ± 1.3	0.7 ± 1.0	2.2 ± 1.2	<0.001
CURB-65 score	1.8 ± 1.1	0.6 ± 0.8	2.0 ± 1.0	<0.001
PSI score	105.9 ± 42.3	45.9 ± 34.9	115.8 ± 34.8	<0.001
I-ROAD score	2.1 ± 0.9	1.2 ± 0.6	2.3 ± 0.8	<0.001
SOFA score	2.7 ± 1.9	1.3 ± 1.1	2.9 ± 1.9	<0.001
Conditions of the patients (mean ± SD)				
SIRS score	0.6 ± 0.5	0.6 ± 0.5	0.6 ± 0.5	0.566
Quick SOFA	0.3 ± 0.5	0.1 ± 0.2	0.3 ± 0.5	<0.001
Bacteremia (n, %) *	26 (11)	1 (4)	25 (13)	0.14
Treatment (n, %)				
ICU admission	15 (4)	3 (5)	12 (3)	0.471
DNAR order	77 (19)	0	77 (22)	<0.001
Mechanical ventilation	19 (5)	0	19 (5)	0.071
Vasopressor use	11 (3)	0	11 (3)	0.174
Initial antibiotic therapy (n, %)				
Penicillin alone	196 (48)	16(28)	180 (52)	0.001
Cephems alone	58 (14)	7 (12)	51 (15)	0.641
Carbapenems alone	70 (17)	7 (12)	63 (18)	0.285
Fluoroquinolones alone	26 (6)	13 (23)	13 (4)	<0.001
Macrolides alone	0	0	0	-
β-lactams plus fluoroquinolones	22 (5)	10 (17)	12 (3)	<0.001
β-lactams plus macrolides	11 (3)	3 (5)	8 (2)	0.2
Others	23 (6)	1 (2)	22 (6)	0.168
Combination plus anti-MRSA agents	5 (1)	0	5 (1)	0.363
Any combination antibiotic therapy	52 (13)	15 (26)	37 (11)	0.001
Antipseudomonal agents use (n, %)	247 (61)	34 (60)	213 (61)	0.843
Route of antibiotics (n, %)				
Oral	7 (2)	5 (9)	2 (1)	0.001
Intravenous	388 (95)	47 (82)	341 (97)	<0.001
Oral and intravenous	12 (3)	5 (9)	7 (2)	0.017
Duration of				
hospital stay (mean days ± SD)	18.6 ± 16.1	12.9 ± 10.2	19.5 ± 16.8	0.004
antibiotics use (mean days ± SD)	13.7 ± 10.8	12.5 ± 9.1	13.9 ± 11.1	0.385
Outcome				
Mortality (n, %)				
30-day mortality	19 (5)	0	19 (5)	<0.001
In-hospital mortality	23 (6)	1 (2)	22 (6)	<0.001
Initial treatment failure (n, %)	37 (9)	5 (9)	32 (9)	0.924
Inappropriate treatment (n, %) **	42 (22)	1 (6)	41 (24)	0.32
Isolating PDR pathogens (n, %)	59 (14)	3 (5)	56 (16)	0.032
Gram positive (n)	***	****	*****	
*Streptococcus pneumoniae*	32 (16.3)	4 (23.5)	28 (15.6)	0.831
*Streptococcus non-pneumonia*	19 (9.7)	0	19 (9.7)	0.381
Methicillin-sensitive *Staphylococcus aureus*	30 (15.3)	1 (5.9)	29 (16.2)	0.083
MRSA	35 (17.9)	0	35 (19.6)	0.013
Coagulase-negative Staphylococci	1 (0.5)	0	1 (0.6)	0.689
*Corynebacterium* species	2 (1)	0	2 (1.1)	0.571
*Enterococcus* species	1 (0.5)	0	1 (0.6)	0.689
Gram-negative (n)	***	****	*****	
*Haemophillus influenzae*	21 (10.7)	6 (35.3)	15 (8.4)	0.042
*Esherichia coli*	18 (9.2)	1 (5.9)	17 (9.5)	0.3
*Pseudomonas aeruginosa*	15 (7.7)	1 (5.9)	14 (7.8)	0.416
*Klebsiella pneumonniae*	26 (13.3)	1 (5.9)	25 (14)	0.422
*Klebsiella oxytoca*	4 (2)	0	4 (2.2)	0.127
*Moraxella catarrahis*	11 (5.6)	2 (11.2)	9 (5)	0.663
*Serratia macescens*	5 (2.6)	1 (5.9)	4 (2.2)	0.682
*Acinetobacter* species	3 (1.5)	1 (5.9)	2 (1.1)	0.319
*Proteus mirabilis*	4 (2)	0	4 (2.2)	0.422
*Stenotrophomonas maltphilia*	2 (1)	0	2 (1.1)	0.573
*Legionella pneumoniae*	1 (0.5)	1 (5.9)	0	0.012
Other Enterobacteriacea †	13 (6.6)	0	13 (7.3)	0.609

DNAR, Do Not Attempt Resuscitation; ICU, intensive care unit; MRSA, methicillin-resistant Staphylococcus aureus; PDR, potential drug resistant; RCT, randomized control trial; SD, standard deviation; SIRS, systemic inflammatory response syndrome; SOFA, sequential organ failure assessment. * These denominators whose blood cultures obtained, are 230, 27 and 203. ** Denominator is 192. Only cases with causative pathogens isolated were analyzed. ***, ****, ***** these denominators whose numbers are positive sputum cultures are 196, 17 and 179. They were calculated up to the first digit of the minority. † contains 3 *Rauotella* sp., 3 *Chryseobacterium* sp., 1 *Pantoea* sp., 1 *Veionella* sp., and 5 *Enterobacter* sp.

**Table 2 jcm-11-00297-t002:** Comparison of patients’ characteristics and outcomes between participation possible and impossible groups among CAP patients.

Variables	All Patients (n = 177)	Participation-Possible Group (n = 51)	Participation-Impossible Group (n = 126)	*p*-Value
Mean age (years ± SD)	71.9 ± 18.3	53.2 ± 17.7	79.5 ± 12.2	<0.001
Median age (years, range)	76 (18–103)	53 (18–79)	82 (37–103)	-
Male gender (n, %)	109 (62)	27 (53)	82 (65)	0.133
Smoking history (n, %)				
Current smoker	26 (15)	11 (22)	15 (12)	0.1
Ex-smoker	82 (46)	20 (39)	62 (49)	0.227
Never smoker	61 (34)	20 (39)	41 (33)	0.397
Unknown	8 (5)	0	8 (6)	0.066
Underlying diseases (n, %)				
Heart disease	51 (40)	4 (8)	47 (37)	<0.001
Chronic pulmonary disease	63 (50)	16 (31)	47 (37)	0.001
Diabetes mellitus	31 (25)	0	31 (25)	<0.001
Chronic kidney disease	14 (11)	0	14 (11)	0.013
Hemodialysis	0	0	0	-
Hepatic disease	4 (3)	0	4 (3)	0.198
Collagen vascular disease	1 (1)	0	1 (1)	0.523
Cerebrovascular disease	28 (22)	0	28 (23)	<0.001
Malignancy	10 (8)	0	10 (8)	0.038
Dementia	23 (13)	1 (2)	22 (17)	0.005
Gastroesophageal reflux disease	4 (3)	2 (4)	2 (2)	0.344
Proton pump inhibitor use	37 (21)	4 (8)	33 (26)	<0.001
Sleep agents use	23 (13)	0	23 (18)	0.001
Charlson comorbidity index (mean ± SD)	1.2 ± 1.1	0.3 ± 0.5	1.6 ± 1.1	<0.001
Charlson comorbidity index ≥ 3 (n, %)	23 (13)	0	23 (18)	0.001
Severity of pneumonia (mean ± SD)				
A-DROP score	1.7 ± 1.2	0.5 ± 0.8	2.1 ± 1.1	<0.001
CURB-65 score	1.5 ± 1.1	0.5 ± 0.7	1.9 ± 1.0	<0.001
PSI score	88.5 ± 44.4	42.5 ± 34.6	107.2 ± 33.1	<0.001
I-ROAD score	1.8 ± 0.9	1.2 ± 0.6	2.1 ± 0.9	<0.001
SOFA score	2.1 ± 1.5	1.3 ± 1.1	2.5 ± 1.5	<0.001
Conditions of the patients (mean ± SD)				
SIRS score	0.6 ± 0.5	0.6 ± 0.5	0.7 ± 0.5	0.208
Quick SOFA	0.2 ± 0.4	0.0 ± 0.2	0.3 ± 0.4	<0.001
Bacteremia (n, %) *	9 (8)	0	9 (11)	0.113
Treatment (n, %)				
ICU admission	6 (3)	3 (6)	3 (2)	0.23
DNAR order	23 (13)	0	23 (18)	0.001
Mechanical ventilation	7 (4)	0	7 (6)	0.086
Vasopressor use	4 (3)	0	4 (3)	0.198
Initial antibiotic therapy (n, %)				
Penicillin alone	70 (40)	11 (22)	59 (47)	0.002
Cephems alone	30 (17)	7 (14)	23 (18)	0.467
Carbapenems alone	26 (15)	7 (14)	19 (15)	0.818
Fluoroquinolones alone	22 (12)	13 (25)	9 (7)	0.001
Macrolides alone	0	0	0	-
β-lactams plus fluoroquinolones	16 (9)	9 (18)	7 (6)	0.011
β-lactams plus macrolides	7 (4)	3 (6)	4 (3)	0.403
Others	6 (3)	1 (2)	5 (4)	0.504
Combination plus anti-MRSA agents	0	0	0	-
Any combination antibiotic therapy	27 (15)	14 (27)	13 (10)	0.004
Antipseudomonal agents use (n, %)	95 (54)	30 (59)	65 (52)	0.382
Route of antibiotics (n, %)				
Oral	5 (3)	5 (10)	0	0.002
Intravenous	167 (94)	42 (82)	125 (99)	<0.001
Oral and intravenous	5 (3)	4 (8)	1 (1)	0.025
Duration of				
hospital stay (mean days ± SD)	16.3 ± 14.7	12.8 ± 9.6	13.6 ± 8.8	0.557
antibiotics use (mean days ± SD)	13.4 ± 9.0	12.4 ± 10.5	17.9 ± 16.0	0.025
Outcome				
Mortality (n, %)				
30-day mortality	3 (2)	0	3 (2)	0.266
In-hospital mortality	5 (3)	1 (2)	4 (3)	0.659
Initial treatment failure (n, %)	10 (6)	4 (8)	6 (5)	0.421
Inappropriate treatment (n, %) **	5 (7)	1 (7)	4 (7)	0.988
Isolating PDR pathogens (n, %)	10 (6)	2 (4)	8 (6)	0.526
Gram positive (n)	***	****	*****	
*Streptococcus pneumoniae*	19 (26.8)	4 (28.6)	15 (26.3)	0.415
*Streptococcus non-pneumonia*	5 (7)	0	5 (8.8)	0.575
Methicillin-sensitive *Staphylococcus aureus*	11 (15.5)	1 (7.1)	10 (17.5)	0.131
MRSA	6 (8.5)	0	6 (10.5)	0.11
Coagulase-negative Staphylococci	0	0	0	-
*Corynebacterium* species	0	0	0	-
*Enterococcus* species	0	0	0	-
Gram-negative (n)				
*Haemophillus influenzae*	12 (16.9)	5 (35.7)	7 (12.3)	0.311
*Esherichia coli*	4 (5.6)	1 (7.1)	3 (5.3)	0.859
*Pseudomonas aeruginosa*	2 (2.8)	1 (7.1)	1 (1.8)	0.509
*Klebsiella pneumonniae*	7 (9.9)	1 (7.1)	6 (10.5)	0.38
*Klebsiella oxytoca*	2 (2.8)	0	2 (3.5)	0.363
*Moraxella catarrahis*	4 (5.6)	1 (7.1)	3 (5.3)	0.859
*Serratia macescens*	2 (2.8)	1 (7.1)	1 (1.8)	0.509
*Acinetobacter* species	0	0	0	-
*Proteus mirabilis*	0	0	1 (1.8)	0.522
*Stenotrophomonas maltphilia*	0	0	0	-
*Legionella pneumoniae*	1 (1.4)	1 (7.1)	0	0.116
*Other Enterobacteriacea* †	7 (9.9)	0	7 (12.3)	0.332

CAP, community-acquired pneumonia; DNAR, Do Not Attempt Resuscitation; ICU, intensive care unit; MRSA, methicillin-resistant Staphylococcus aureus; PDR, potential drug resistant; SD, standard deviation; SIRS, systemic inflammatory response syndrome; SOFA, sequential organ failure assessment. * Patients who obtained a blood culture were evaluated. Then, the denominators are 107, 25, and 82 in all patients, the RCT appropriate group and RCT inappropriate group, respectively. ** Patients who had causative pathogens identified were evaluated. Then, the denominators were 70, 15, and 55 in all patients in the RCT appropriate group and RCT inappropriate group, respectively. ***, ****, ***** These denominators, whose number are positive sputum cultures, are 71, 14, and 57, respectively. They were calculated up to the first digit of the minority. † contains 1 *Pantoea* sp., 1 *Chryseobacterium* sp., 1 *Veionella* sp., 1 *Rauotella* sp., and 3 *Enterobacter* sp.

**Table 3 jcm-11-00297-t003:** Comparison of patients’ characteristics and outcomes between participation possible and impossible groups among HCAP patients.

Variables	All Patients (n = 229)	Participation-Possible Group (n = 6)	Participation-Impossible Group (n = 223)	*p*-Value
Mean age (years ± SD)	78.1 ± 10.6	69.5 ± 6.4	78.4 ± 10.6	0.304
Median age (years, range)	80 (42–99)	69 (62–78)	80 (42–99)	-
Male gender (n, %)	148 (65)	1 (17)	147 (66)	0.013
Smoking history (n, %)				
Current smoker	0	0	10 (4)	0.596
Ex-smoker	123 (54)	4 (67)	119 (53)	0.519
Never smoker	74 (32)	1 (17)	73 (33)	0.406
Unknown	22 (10)	1 (17)	21 (9)	0.552
Underlying diseases (n, %)				
Heart disease	75 (33)	0	75 (34)	0.083
Chronic pulmonary disease	112 (49)	4 (67)	108 (48)	0.378
Diabetes mellitus	29 (13)	0	29 (13)	0.345
Chronic kidney disease	37 (16)	0	37 (17)	0.276
Hemodialysis	15 (7)	0	15 (7)	0.511
Hepatic disease	11 (5)	0	11 (5)	0.577
Collagen vascular disease	40 (17)	0	40 (18)	0.253
Cerebrovascular disease	72 (31)	0	72 (32)	0.093
Malignancy	65 (28)	0	65 (29)	0.118
Dementia	51 (22)	1 (17)	50 (22)	0.738
Gastroesophageal reflux disease	10 (4)	1 (17)	11 (5)	0.135
Proton pump inhibitor use	85 (37)	1 (17)	84 (38)	0.293
Sleep agents use	37 (16)	0	37 (17)	0.273
Charlson comorbidity index (mean ± SD)	2.7 ± 2.0	0.8 ± 0.4	2.8 ± 2.1	0.022
Charlson comorbidity index ≥ 3 (n, %)	98 (43)	0	98 (44)	0.032
Severity of pneumonia (mean ± SD)				
A-DROP score	2.3 ± 1.3	1.8 ± 1.8	2.3 ± 1.2	0.154
CURB-65 score	2.1 ± 1.0	1.5 ± 0.8	2.1 ± 1.0	0.648
PSI score	119.4 ± 35.2	75.0 ± 23.3	120.6 ± 34.8	0.219
I-ROAD score	2.3 ± 0.8	1.5 ± 0.8	2.4 ± 0.8	0.685
SOFA score	3.2 ± 2.1	1.7 ± 1.4	3.2 ± 2.2	0.193
Conditions of the patients (mean ± SD)				
SIRS score	0.6 ± 0.5	0.7 ± 0.5	0.6 ± 0.5	0.162
Quick SOFA	1.2 ± 0.8	0.2 ± 0.4	0.4 ± 0.5	0.001
Bacteremia (n, %) *	17 (14)	1 (50)	16 (13)	0.26
Treatment (n, %)				
ICU admission	9 (4)	0	9 (4)	0.615
DNAR order	54 (24)	0	54 (24)	0.168
Mechanical ventilation	12 (5)	0	12 (5)	0.559
Vasopressor use	7 (3)	0	7 (3)	0.659
Initial antibiotic therapy (n, %)				
Penicillin alone	126 (55)	5 (83)	121 (54)	0.158
Cephems alone	28 (12)	0	28 (13)	0.354
Carbapenems alone	44 (19)	0	44 (20)	0.226
Fluoroquinolones alone	4 (2)	0	4 (2)	0.741
Macrolides alone	0	0	0	-
β-lactams plus fluoroquinolones	6 (3)	1 (17)	5 (2)	0.029
β-lactams plus macrolides	4 (2)	0	4 (2)	0.741
Others	17 (7)	0	17 (8)	0.482
Combination plus anti-MRSA agents	5 (2)	0	5 (2)	0.711
Any combination antibiotic therapy	25 (11)	1 (17)	24 (11)	0.647
Antipseudomonal agents use (n, %)	152 (66)	4 (67)	148 (66	0.998
Route of antibiotics (n, %)				
Oral	2 (1)	0	2 (1)	1.000
Intravenous	220 (96)	5 (83)	215 (96)	0.216
Oral and intravenous	7 (3)	1 (17)	6 (3)	0.172
Duration of				
hospital stay (mean days ± SD)	20.4 ± 16.9	16.7 ± 7.4	20.5 ± 17.2	0.284
antibiotics use (mean days ± SD)	14.0 ± 12.0	10.8 ± 4.0	14.0 ± 12.2	0.349
Outcome				
Mortality (n, %)				
30-day mortality	16 (5)	0	16 (7)	0.456
In-hospital mortality	18 (6)	0	18 (8)	0.468
Initial treatment failure (n, %)	27 (9)	1 (17)	26 (12)	0.924
Inappropriate treatment (n, %) **	37 (31)	0	37 (32)	0.559
Isolating PDR pathogens (n, %)	49 (14)	1 (17)	48 (22)	0.775
Gram positive (n)	***	****	*****	
*Streptococcus pneumoniae*	13 (10.4)	0	13 (10.7)	0.584
*Streptococcus non-pneumonia*	14 (11.2)	0	14 (11.5)	0.571
Methicillin-sensitive *Staphylococcus aureus*	19 (15.2)	0	19 (15.6)	0.5
MRSA	29 (23.2)	0	29 (23.8)	0.391
Coagulase-negative Staphylococci	1 (0.8)	0	1 (0.8)	0.883
*Corynebacterium* species	2 (1.6)	0	2 (1.6)	0.835
*Enterococcus* species	1 (0.8)	0	1 (0.8)	0.883
Gram-negative (n)	***	****	*****	
*Haemophillus influenzae*	9 (7.2)	1 (35.3)	8 (6.6)	0.055
*Esherichia coli*	14 (11.2)	0	14 (11.5)	0.569
*Pseudomonas aeruginosa*	13 (10.4)	0	13 (10.7)	0.584
*Klebsiella pneumonniae*	19 (15.2)	0	19 (15.6)	0.5
*Klebsiella oxytoca*	2 (1.6)	0	2 (1.6)	0.835
*Moraxella catarrahis*	7 (5.6)	1 (11.2)	6 (4.9)	0.022
*Serratia macescens*	3 (2.4)	0	3 (2.5)	0.798
*Acinetobacter* species	3 (2.4)	1 (5.9)	2 (1.6)	<0.001
*Proteus mirabilis*	3 (2.4)	0	3 (2.5)	0.798
*Stenotrophomonas maltphilia*	2 (1.6)	0	2 (1.6)	0.836
*Legionella pneumoniae*	0	0	0	-
*Other Enterobacteriacea* †	6 (4.8)	0	6 (4.9)	0.717

DNAR, Do Not Attempt Resuscitation; HCAP, healthcare-associated pneumonia; ICU, intensive care unit; MRSA, methicillin-resistant Staphylococcus aureus; PDR, potential drug resistant; RCT, randomized control trial; SD, standard deviation; SIRS, systemic inflammatory response syndrome; SOFA, sequential organ failure assessment. * Patients who obtained a blood culture were evaluated. Then, the denominators were 122, 2, and 120 in all patients in the RCT appropriate group and RCT inappropriate group, respectively. ** Patients who had causative pathogens identified were evaluated. Then, the denominators were 118, 3, and 115 in all patients in the RCT appropriate group and RCT inappropriate group, respectively. ***, ****, ***** These denominators, whose number are positive sputum cultures, are 125, 3, and 122, respectively. They were calculated up to the first digit of the minority. † contains 2 *Rauotella* sp., 2 *Chryseobacterium* sp., and 2 *Enterobacter* sp.

**Table 4 jcm-11-00297-t004:** Reasons the patients are not eligible for clinical trials (n = 349).

Factors	n (%)
1. Age (<18, >80 years old)	180 (52)
2. Underlying disease which could not be assessed	254 (73)
Heart disease	106 (30)
Pulmonary disease	74 (21)
Kidney disease	37 (11)
Hepatic disease	15 (4)
Cerebrovascular disease	19 (5)
Diabetes mellitus	39 (12)
Collagen vascular disease	41 (12)
Malignancy	63 (18)
Mental disorder	12 (4)
3. Aspiration pneumonia	196 (56)
4. Immunosuppressor agents use ^Ж^	44 (13)
5. Chemotherapy	25 (7)
6. Hemodialysis	18 (5)
7. Poor ADL or required any help	
ECOG-PS ≥ 3	111 (32)
Tube feeding	20 (6)
Home oxygen therapy	28 (8)
8. Other infections complicated	11 (3)
9. Requiring mechanical ventilation and/or ICU admission	20 (6)
10. Poor life expectancy	20 (6)
11. Pregnancy	0

ADL, activities of daily living; ECOG-PS, Eastern Cooperative Oncology Group performance status; ICU, intensive care unit. ^Ж^ corticosteroids included.

## Data Availability

All data generated or analyzed during this study are included in this published article.

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
