# Peer review of "A Large Gap in Patients’ Characteristics and Outcomes between the Real-World and Clinical Trial Settings in Community-Acquired Pneumonia and Healthcare-Associated Pneumonia"

_jcm, 2022, doi:10.3390/jcm11020297_

Round 1
Reviewer 1 Report
The topic addressed by this study is innovative and seems to me of interest to the clinician. To what extent are patients included in randomized trials representative of real life?
The design of the study is appropriate.
I would like to suggest the following modifications:
Methods.
Study design : please also include ICU admission and DNAR orders in the variables of interest listed.
How were data lacking managed ?
"We reviewed all community-onset pneumonia ?" What about HCAP ?
Microbiologial evaluation : "A sputum sample… was collected" Had really all patients a sputum sample ? In our experience, many older / frail patients cannot spit..
Statistical analysis :
Please also present Survival curves / log rank test.
Results :
Please be consistent between the text and the table and use the same terms int the whole manuscrit ("participation possible" or "RCT appropriate")
"MRSA is more frequently seen in participation impossible group than in participation impossible group" : please modify.
Tables : Please methicillin-sensitive staphylococcus
Please specify other Enterobacteriaceae, as E coli, K; pneumoniae, etc.. are Enterobacteriaceae.
there are many typing errors in the names of bacteria (text and tables), please check meticulously in the whole manuscript.
Discussion :
1- The study concluded that randomized trials represent an unrepresentative proportion of the population. One line of discussion could be the importance of observational studies in which the exclusion/non-inclusion criteria are less strict and the study population more representative : Medical research cannot be limited to data from randomized trials, as observational data are complementary to interventional studies conducted on selected populations.
2- I would also suggest addressing the importance of randomized trials specifically dedicated to the oldest/fragile people, who are usually excluded from these trials.
Limits :
1- The monocentric nature of the study is also an important limitation in terms of the extrapolability of the results.
2- The fact that the patients included in this study correspond to the international diagnostic criteria is in itself a selection bias: in clinical practice, there are obviously pneumonias that do not meet these research criteria, particularly in the frail elderly patient with an atypical clinical presentation.
Author Response
Manuscript ID: jcm-1483922      Date: Dec 26, 2021
Title: A large gap of patients' characteristics and outcomes between the real-world and clinical trial settings in community-onset pneumonia
Thank you very much for your great reviewing. We totally agree with your opinions and made appropriate revisions according to your suggestions. The revised portion was red in the text. We strongly believe that this revision made the text better than before. Please, consider my article for publication in your journal.
Sincerely,
Nobuhiro Asai and Hiroshige Mikamo,

Reviewer 2 Report
In this manuscript, Asai and colleagues report on a retrospective analysis on the eligibility of their pneumonia patients in a potential RCT. They found that a very small proportion of the patients that they treated at their institution would have been eligible for enrollment in clinical trials, postulating important implications.
Overall, I think that the manuscript is interesting in terms of the surrounding idea behind the work. An analysis on how many patients would be eligible for inclusion in an RCT is obviously interesting, and I think that represents the major strength point of this piece.
However, there are several issues that should be addressed prior to formulate a final judgment on this manuscript.
In details:
1) The authors reported about the eligibility in a "clinical trial", without specifying what kind of clinical trial they are referring to. This is an important information since inclusion/exclusion criteria may change according to the aim of the RCT, so a more specific focus should be proposed: are they looking for the eligibility rate for a potential RCT investigating a new antibiotic? Or a ventilation technique? Or perhaps oxygen saturation target? This should be specify, the incl/excl criteria adjusted, and the analysis performed accordingly.
2) Relatedly, the overarching aim of the mansucript should be reported more clearly in the introduction of the manuscript, in order to help the reader focus on the point of this analysis.
3) Again, the authors should specify what kind of literature search was done to identify the incl/excl criteria against to which their population was screened for. How were these selected? The authors stated that the exclusion criteria reported were "commonly used in the many past ordinary clinical trials" but this statement is too generic and more specific reference should be reported. See also comment #1, which is related to this one.
4) One important point is the definition of the population. The authors stated in the title that they focused on "community-onset pneumonia", but they also included the so-called HCAP patients (healthcare acquired pneumonia) in their analysis. This is misleading and should be avoided, since a trial would be focused on one of these two population. Perhaps the authors want to update their analysis (which is stucked at 2017) to increase the sample size of CAP patients, if they think that would help increase the power of the analysis.
5) Related to point 4), please note that the definition of HCAP has become obsolete after further updates of the international guidelines (10.1093/cid/ciw353).
6) There are some mistakes in the reporting of the results in the text, which inverted the meaning of some sentences (see for example "PDR pathogens were seen more frequently in the participation possible group than in
those within the participation impossible group" - while in the table is the opposite; similarly, page 17, "Outstandingly, the OSs in the participation impossible group with CAP were significantly shorter OSs than those in the participation the impossible group" - the meaning of the sentence is the opposite of the data showed".
7) As a general comment, the English language and grammar needs significant improvement - there are some passages that are difficult to understand. As a complimentary example see the heading of Table 4: "Reasons of clinical trials of which is inappropriate (n=349)." - the sentence is difficult to understand.
8) How about steroids therapy? Increasing report deals with potential harm and benefits of steroids in the pneumonia setting (e.g. 0.1097/MD.0000000000014636; 10.1001/jama.2020.0216; 10.1513/AnnalsATS.201806-419OC).
9) Finally, a more extensive discussion of the findings presented should be provided, specifically in terms of a) clinical implication of the results presented: how can these low rate of eligibility in RCT can slow research on the pneumonia topic? b) on the addition of this specific work on the current evidence - how this paper fit into current literature, what kind of information/value does it add?; c) further research and clinical scenarios - what kind of strategies can be implemented to increase the eligibility rate of these patients in clinical trials? what are the major obstacles? d) how did (and will) Covid-19 affect all these aspect, and interefere with potential eligibility (especially in the screening phase) in clinical trials? all these (and others) aspects should be covered more extensively (and referenced adequately) in the discussion section.
Author Response
Manuscript ID: jcm-1483922      Date: Dec 26, 2021
Title: A large gap of patients' characteristics and outcomes between the real-world and clinical trial settings in community-onset pneumonia
Thank you very much for your great reviewing. We totally agree with your opinions and made appropriate revisions according to your suggestions. The revised portion was red in the text. We strongly believe that this revision made the text better than before. Please, consider my article for publication in your journal.
Sincerely,
Nobuhiro Asai and Hiroshige Mikamo,
Reviewer 2
Comments and Suggestions for Authors
In this manuscript, Asai and colleagues report on a retrospective analysis on the eligibility of their pneumonia patients in a potential RCT. They found that a very small proportion of the patients that they treated at their institution would have been eligible for enrollment in clinical trials, postulating important implications.
Overall, I think that the manuscript is interesting in terms of the surrounding idea behind the work. An analysis on how many patients would be eligible for inclusion in an RCT is obviously interesting, and I think that represents the major strength point of this piece.
However, there are several issues that should be addressed prior to formulate a final judgment on this manuscript.
In details:
1) The authors reported about the eligibility in a "clinical trial", without specifying what kind of clinical trial they are referring to. This is an important information since inclusion/exclusion criteria may change according to the aim of the RCT, so a more specific focus should be proposed: are they looking for the eligibility rate for a potential RCT investigating a new antibiotic? Or a ventilation technique? Or perhaps oxygen saturation target? This should be specify, the incl/excl criteria adjusted, and the analysis performed accordingly.
2) Relatedly, the overarching aim of the mansucript should be reported more clearly in the introduction of the manuscript, in order to help the reader focus on the point of this analysis.
3) Again, the authors should specify what kind of literature search was done to identify the incl/excl criteria against to which their population was screened for. How were these selected? The authors stated that the exclusion criteria reported were "commonly used in the many past ordinary clinical trials" but this statement is too generic and more specific reference should be reported. See also comment #1, which is related to this one.
Answer to 1)-3)
I agree with your helpful and informative suggestions. Therefore, I added some sentences as follows. Could you accept my revisions as follows?
- Introduction
Evidence-based medicine (EBM) aims to assist physicians in making rational decisions in general practices. As EBM is established according to the results of clinical trials, clinical trials are considered to be one of the most important undertakings and are put at the top of priority among physicians in constructing therapeutic strategies [1]. The efficacy and tolerability of a new antibiotic treatment is evaluated by a randomized control trial and EBM guidelines/recommendations are made based on those results. There is no room for doubt that current medicine is based on EBM. However, we skeptically think about that when we consider ​eligibility of pneumonia patients for EBM guidelines/recommendations in actual practice. Entry criteria for any clinical trial are generally very strict, and most patients might not be suitable for the studies. Thus, it is reasonable to doubt if the results of clinical trials reflect the real world in general practice. We already reported that only 24 % of candidemia patients could be ​eligible in a clinical trial [2]. Pneumonia remains ​a leading cause of infection deaths worldwide [3,4]. Especially, elderly patients with pneumonia tend to have more comorbidities than young patients and the mortality rate is higher than other groups [4,5]. We have suspected that there might be a distinct difference of clinical pictures (characteristics) between the patients eligible and those excluded from the study and decided to perform this study. This study focused on to what degree of community-acquired pneumonia (CAP) and healthcare-associated pneumonia (HCAP)community-onset pneumonia patients are eligible for clinical trials to investigate whether antibiotic therapy is effective and/or tolerable for the patients. This is the first report demonstrating to what degree clinical data on which EBM is based ​on, reflects the real world patients with pneumonia.
- Methods
2.1. Study design
Our institute is a 900-bed tertiary care center and is located in the countryside at Aichi prefecture in central Japan. For the purpose of how many community-onset pneumonia patients in our institute could join any randomized clinical trials for an antibiotic treatment among pneumonia patients, we reviewed all CAP and HCAP patients who were admitted in our hospital between September 2014 and May 2017. Pneumonia was diagnosed according to the previously published international guidelines [6]. CAP and HCAP were categorized based on the criteria published by the American Thoracic Society/ Infectious Diseases Society of America (ATS/IDSA) in 2006 [7,8]. Severity of pneumonia was evaluated by A-DROP [8], CURB-65 [9], Pneumonia Severity Index (PSI) [10], I-ROAD [11] and SOFA score [12]. Comorbidity was evaluated by the Charlson comorbidity index (CCI) [13]. The patients were divided into two groups: patients who were eligible for clinical trials (participation possible group), and those who were not (participation impossible group). Then, patients’ characteristics (age, sex), pathogens isolated, clinical outcomes such as the treatments, 30-day, or in-hospital mortality and the reasons of exclusion from the clinical trial were evaluated.
4) One important point is the definition of the population. The authors stated in the title that they focused on "community-onset pneumonia", but they also included the so-called HCAP patients (healthcare acquired pneumonia) in their analysis. This is misleading and should be avoided, since a trial would be focused on one of these two population. Perhaps the authors want to update their analysis (which is stucked at 2017) to increase the sample size of CAP patients, if they think that would help increase the power of the analysis.
A). I replaced the word ”community-onset pneumonia” to CAP and HCAP as follows because other review suggested the same point. I revised the title, too.
- Methods
2.1. Study design
Our institute is a 900-bed tertiary care center and is located in the countryside at Aichi prefecture in central Japan. For the purpose of how many community-onset pneumonia patients in our institute could join any randomized clinical trials for an antibiotic treatment among pneumonia patients, we reviewed all CAP and HCAP patients who were admitted in our hospital between September 2014 and May 2017. Pneumonia was diagnosed according to the previously published international guidelines [6]. CAP and HCAP were categorized based on the criteria published by the American Thoracic Society/ Infectious Diseases Society of America (ATS/IDSA) in 2006 [7,8]. Severity of pneumonia was evaluated by A-DROP [8], CURB-65 [9], Pneumonia Severity Index (PSI) [10], I-ROAD [11] and SOFA score [12]. Comorbidity was evaluated by the Charlson comorbidity index (CCI) [13]. The patients were divided into two groups: patients who were eligible for clinical trials (participation possible group), and those who were not (participation impossible group). Then, patients’ characteristics (age, sex), pathogens isolated, clinical outcomes such as the treatments, 30-day, or in-hospital mortality and the reasons of exclusion from the clinical trial were evaluated.
5) Related to point 4), please note that the definition of HCAP has become obsolete after further updates of the international guidelines (10.1093/cid/ciw353).
A). Yes, your suggestion is very important. HCAP has been removed from the HAP/VAP guidelines. The main reason is that contact with the health care system is not a strong predictor of risk for MDR bacteria. And microbial profiles of HCAP are similar to the one of CAP. However, patients’ profile in HCAP is different from the one in CAP in Japan. Therefore, concept of HCAP changed (It is called as NHCAP in Japan.)
Then, I made some revision on the text as follows.
In the Introduction
- Introduction
Evidence-based medicine (EBM) aims to assist physicians in making rational decisions in general practices. As EBM is established according to the results of clinical trials, clinical trials are considered to be one of the most important undertakings and are put at the top of priority among physicians in constructing therapeutic strategies [1]. The efficacy and tolerability of a new antibiotic treatment is evaluated by a randomized control trial and EBM guidelines/recommendations are made based on those results. There is no room for doubt that current medicine is based on EBM. However, we skeptically think about that when we consider ​eligibility of pneumonia patients for EBM guidelines/recommendations in actual practice. Entry criteria for any clinical trial are generally very strict, and most patients might not be suitable for the studies. Thus, it is reasonable to doubt if the results of clinical trials reflect the real world in general practice. We already reported that only 24 % of candidemia patients could be ​eligible in a clinical trial [2]. Pneumonia remains ​a leading cause of infection deaths worldwide [3,4]. Especially, elderly patients with pneumonia tend to have more comorbidities than young patients and the mortality rate is higher than other groups [4,5]. Since it was found that contact with the health care facility is not a strong predictor of risk for multidrug-resistant bacteria, healthcare-associated pneumonia (HCAP) has been removed from hospital-acquired pneumonia (HAP)/ventilator-associated pneumonia guidelines. However, HCAP in Japan was included in HAP due to the greater patients’ profile diversity of HCAP than CAP [6,7]. We have suspected that there might be a distinct difference of clinical pictures (characteristics) between the patients eligible and those excluded from the study and decided to perform this study. This study focused on to what degree of community-acquired pneumonia (CAP) and (HCAP) patients are eligible for clinical trials to investigate whether antibiotic therapy is effective and/or tolerable for the patients. This is the first report demonstrating to what degree clinical data on which EBM is based ​on, reflects the real-world patients with pneumonia.
6) There are some mistakes in the reporting of the results in the text, which inverted the meaning of some sentences (see for example "PDR pathogens were seen more frequently in the participation possible group than in
those within the participation impossible group" - while in the table is the opposite; similarly, page 17, "Outstandingly, the OSs in the participation impossible group with CAP were significantly shorter OSs than those in the participation the impossible group" - the meaning of the sentence is the opposite of the data showed".
A). I am so sorry to confuse you. It’s my fault and I revised the sentences as follows. As you suggest, all of them were opposite. Thank you so much for noticing us those mistakes. I appreciate it.
In page 3 of 20
- Discussion
Patients in the real world are quite different from those who can participate in a clinical trial. We already reported that only 24% of candidemia patients could participate in a clinical trial. Patients who can participate in a clinical trial have better PSs and longer overall survival times than those seen in actual medical practice [2]. Like the study, community-onset pneumonia patients with participation possible group showed lesser severity of pneumonia and fewer comorbidities than those with the participation impossible group. We found the participation impossible group had higher mortality rates of 30-day and in-hospital than the participation possible group. However, identification of PDR pathogens, mechanical ventilation and Do Not Attempt Resuscitation (DNAR) order were more frequently seen in the participation impossible group than in the possible group. In Japan, discussing DNAR order with Japanese family members is still considered to be taboo [25].
7) As a general comment, the English language and grammar needs significant improvement - there are some passages that are difficult to understand. As a complimentary example see the heading of Table 4: "Reasons of clinical trials of which is inappropriate (n=349)." - the sentence is difficult to understand.
A). Thank you for your suggestion. The sentence was revised as follows. I hope it makes sense.
Table 4. Reasons the patients who are not eligible for clinical trials
8) How about steroids therapy? Increasing report deals with potential harm and benefits of steroids in the pneumonia setting (e.g. 0.1097/MD.0000000000014636; 10.1001/jama.2020.0216; 10.1513/AnnalsATS.201806-419OC).
A). I figure out what you suggest. For several clinical trials such as pneumonia, malignancy, cases using corticosteroid are excluded because it is hard to evaluate whether a targeted treatment are effective for the cases. Then, we added the sentence as follows.
In Table 4
Ж included corticosteroids.
9) Finally, a more extensive discussion of the findings presented should be provided, specifically in terms of a) clinical implication of the results presented: how can these low rate of eligibility in RCT can slow research on the pneumonia topic? b) on the addition of this specific work on the current evidence - how this paper fit into current literature, what kind of information/value does it add?; c) further research and clinical scenarios - what kind of strategies can be implemented to increase the eligibility rate of these patients in clinical trials? what are the major obstacles? d) how did (and will) Covid-19 affect all these aspect, and interefere with potential eligibility (especially in the screening phase) in clinical trials? all these (and others) aspects should be covered more extensively (and referenced adequately) in the discussion section.
Answer to 9)
Yes. As you mentioned, I think that the discussion is not good enough to support our results and what the point of article is. Then, I added some sentences as follows according to your suggestions.
My points are two major things. One is that “We should consider a randomized control trial focusing on the elderly or fragile people who are usually excluded from the trials or analyze alternatives such as propensity score matching analysis. These will be helpful for clinicians to make a rational decision in the treatment among those people.” The other is that “Though EBM is very important and essential to advance medicine, we should acknowledge the facts and limits of clinical trials. Every physician should not be overconfident in EBM based on the results of a clinical trial.” which is written in Conclusion.
In Discussion
- Discussion
Patients in the real world are quite different from those who can participate in a clinical trial. We already reported that only 24% of candidemia patients could participate in a clinical trial. Patients who can participate in a clinical trial have better PSs and longer overall survival times than those seen in actual medical practice [2]. Like the study, community-onset pneumonia patients with participation possible group showed lesser severity of pneumonia and fewer comorbidities than those with the participation impossible group. We found the participation impossible group had higher mortality rates of 30-day and in-hospital than the participation possible group. However, identification of PDR pathogens, mechanical ventilation and Do Not Attempt Resuscitation (DNAR) order were more frequently seen in the participation impossible group than in the possible group. In Japan, discussing DNAR order with Japanese family members is still considered to be taboo [25]. Therefore, these results could suggest that patients in the participation impossible group have a worse prognosis than those in the participation possible group do. It is well known that HCAP patients are more likely to have worse PSs and more comorbidities than those with CAP [4,5,7]. We should consider a randomized control trial focusing on the elderly or fragile people who are usually excluded from the trials or analyze alternatives such as propensity score matching analysis. These will be helpful for clinicians to make a rational decision in the treatment among those people.
Outstandingly, the OSs in the participation impossible group with CAP were significantly shorter OSs than those in the participation possible group, while 30-day and in-hospital mortality rate did not differ between the two groups. More comorbidities could affect the prognosis among the participation impossible group. Particularly, more aspiration pneumonia was seen in 65/125 (52%) and 140/223 (63%) among participation impossible group with CAP and HCAP, respectively. Performance status in patients with aspiration pneumonia are likely to decline, some of them become bedridden [26]. These poor conditions can lead to a lower survival rate in participation impossible group. Unfortunately, we did not analyze these data. Physicians should pay attention to them after discharge. Besides, 97% of HCAP patients in the studies [4,5,7] was excluded from the clinical trial. In addition, HCAP patients in the participation possible group had much shorted durations of antibiotic treatment and admission than those in the participation impossible group. An appropriate duration of antibiotics is said to be 5-7 days. A sub-analysis showed that there was no difference of mean duration of antibiotic therapy between survival and 30-day death group among HCAP patients (survival 14.2 vs. 30-day death 10.3 days, p=0.21). The result of our study also suggests that HCAP patients are likely to have longer duration of antibiotic therapy 10-14 days as we expected. The therapeutic strategy for HCAP patients might have to be reconsidered due to the poor general conditions.
As for an initial antibiotic therapy among CAP patients, more penicillins and less fluoroquinolones were seen in the participation impossible group than the participation possible group. The reasons are the initial antibiotic selections were based on the patients’ characteristics. The patients who received penicillins were aspiration pneumonia in 31/70 (44.3%) and those who received fluoroquinolones were younger than 50 years in 7/21 (33.3%). The doctors prescribed penicillins and fluoroquinolones for the patients to cover anerobic bacteria and atypical bacteria, respectively.
There are several limitations in our study. First, this is a retrospective study in a small population. Thus, there might be a bias in data selection and analysis such as the severity of pneumonia, Second, we evaluated only patients who were admitted to our institute. Choice of an initial antibiotic therapy, indication of hospitalization, ICU admission and DNAR orders were based on the physicians’ decision. There might be possibility that patients in this study could not reflect the whole ​of pneumonia patients.

Reviewer 3 Report
The authors noted a large discrepancy between the backgrounds of pneumonia patients in the real world and patients who are enrolled in clinical trials. And to clarify the difference, they examined the difference between a group of 406 consecutive patients with pneumonia who could be enrolled in clinical trials and a group of patients who could not. I agree that the difference between clinical trials and real world in patients with bacterial pneumonia that the authors focused on is a very important point. However, I feel that there are many things that need to be improved in this paper.
- The target group for this study is hospitalized patients. It is necessary to clarify what the indications for hospitalization were based on.
- The Table describes the antibiotics that were used for the first therapy. The criteria for the choice of antibiotics should also be described. It also should be consider about route of antibiotics, intravenous or oral.
- The antimicrobial agents selected tended to be more Penicillins and less Fluoroquinolones in the RCT impossible group. Shouldn't you add some discussion about the cause of this?
- Although the duration of hospitalization was significantly longer in the RCT impossible group compared to the RCT possible group, there was no difference in the duration of antimicrobial treatment. The reason for this should also be considered.
- Please recheck all the data in the Table to make sure it is correct. There are a few problems I wondered: Smoking history in Table 2 is listed as 0 patients (2%) in the RCT possible group. Is this correct that the duration of hospitalization in Table 2 is shorter than the duration of antibiotics? Please be consistent in whether percentages should be written as integers or to one decimal place.
- In the text, RCT possible group is mentioned, but in the Table, RCT appropriate group is used. Please standardize the terms.
- I think the overall survival data shown in the figures is important, but there is no discussion of it, so it is difficult to understand what meaning this OS data has in this study. Please add a discussion of overall survival. You should also add the p-value to the figure.
- The Discussion describes that the duration of antibiotics for HCAP seems to be good at 5-7 days from this study. Why is this? The duration of antibiotics for HCAP in this study is about 10 days, and there are only 6 patients in the RCT possible group for HCAP. I feel that this is not an appropriate consideration.
- The Conclusion should also be reevaluated, I believe. Is it the only important point of this study that only 14% of pneumonia patients are eligible to participate in clinical trials? Isn't the important point that many patients with poor physical status, much complications or severe illness are not considered in the clinical trials?
Author Response
Manuscript ID: jcm-1483922      Date: Dec 26, 2021
Title: A large gap of patients' characteristics and outcomes between the real-world and clinical trial settings in community-onset pneumonia
Thank you very much for your great reviewing. We totally agree with your opinions and made appropriate revisions according to your suggestions. The revised portion was red in the text. We strongly believe that this revision made the text better than before. Please, consider my article for publication in your journal.
Sincerely,
Nobuhiro Asai and Hiroshige Mikamo,
Reviewer 3
Comments and Suggestions for Authors
The authors noted a large discrepancy between the backgrounds of pneumonia patients in the real world and patients who are enrolled in clinical trials. And to clarify the difference, they examined the difference between a group of 406 consecutive patients with pneumonia who could be enrolled in clinical trials and a group of patients who could not. I agree that the difference between clinical trials and real world in patients with bacterial pneumonia that the authors focused on is a very important point. However, I feel that there are many things that need to be improved in this paper.
- The target group for this study is hospitalized patients. It is necessary to clarify what the indications for hospitalization were based on.
A). Yes, I think it's necessary. But, there might have been an issue that patients' admission criteria are different country by country. We have no definite criteria for patients' admission at our institute. Then, I added some sentences as follows.
In page 4 of 19 before conclusion.
There are several limitations in our study. First, this is a retrospective study in a small population. Thus, there might be a bias in data selection and analysis such as the severity of pneumonia, Second, we evaluated only patients who were admitted to our institute. Choice of an initial antibiotic therapy, indication of hospitalization, ICU admission and DNAR orders were based on the physicians’ decision. There might be possibility that patients in this study could not reflect the whole ​of pneumonia patients.
- The Table describes the antibiotics that were used for the first therapy. The criteria for the choice of antibiotics should also be described. It also should be consider about route of antibiotics, intravenous or oral.
A). I agree with your opinion and added these information about the administration route of antibiotics into the
tables.
- The antimicrobial agents selected tended to be more Penicillins and less Fluoroquinolones in the RCT impossible group. Shouldn't you add some discussion about the cause of this?
A). I agree with your opinion. Then, I added the following sentences.
In page 3 of 19
As for an initial antibiotic therapy among CAP patients, more penicillins and less fluoroquinolones were seen in the participation impossible group than the participation possible group. The reasons are the initial antibiotic selections were based on the patients’ characteristics. The patients who received penicillins were aspiration pneumonia in 31/70 (44.3%) and those who received fluoroquinolones were younger than 50 years in 7/21 (33.3%). The doctors prescribed penicillins and fluoroquinolones for the patients to cover anerobic bacteria and atypical bacteria, respectively.
- Although the duration of hospitalization was significantly longer in the RCT impossible group compared to the RCT possible group, there was no difference in the duration of antimicrobial treatment. The reason for this should also be considered.
A). I agree with your opinion, then, the following sentences were added.
In page 3 of 19
Outstandingly, the OSs in the participation impossible group with CAP were significantly shorter OSs than those in the participation possible group, while 30-day and in-hospital mortality rate did not differ between the two groups. More comorbidities could affect the prognosis among the participation impossible group. Particularly, more aspiration pneumonia was seen in 65/125 (52%) and 140/223 (63%) among participation impossible group with CAP and HCAP, respectively. Performance status in patients with aspiration pneumonia are likely to decline, some of them become bedridden [26]. These poor conditions can lead to a lower survival rate in participation impossible group. Unfortunately, we did not analyze these data.
- Please recheck all the data in the Table to make sure it is correct. There are a few problems I wondered: Smoking history in Table 2 is listed as 0 patients (2%) in the RCT possible group. Is this correct that the duration of hospitalization in Table 2 is shorter than the duration of antibiotics? Please be consistent in whether percentages should be written as integers or to one decimal place.
A). It’s totally my fault. I am so sorry that I overlooked it. The table was revised correctly.
- In the text, RCT possible group is mentioned, but in the Table, RCT appropriate group is used. Please standardize the terms.
A). It is a simple mistake. I revised them correctly.
- I think the overall survival data shown in the figures is important, but there is no discussion of it, so it is difficult to understand what meaning this OS data has in this study. Please add a discussion of overall survival. You should also add the p-value to the figure.
A). I agree with your opinion, then, the following sentences were added just the same as above.
In page 3 of 19
Outstandingly, the OSs in the participation impossible group with CAP were significantly shorter OSs than those in the participation possible group, while 30-day and in-hospital mortality rate did not differ between the two groups. More comorbidities could affect the prognosis among the participation impossible group. Particularly, more aspiration pneumonia was seen in 65/125 (52%) and 140/223 (63%) among participation impossible group with CAP and HCAP, respectively. Performance status in patients with aspiration pneumonia are likely to decline, some of them become bedridden [26]. These poor conditions can lead to a lower survival rate in participation impossible group. Unfortunately, we did not analyze these data.
- The Discussion describes that the duration of antibiotics for HCAP seems to be good at 5-7 days from this study. Why is this? The duration of antibiotics for HCAP in this study is about 10 days, and there are only 6 patients in the RCT possible group for HCAP. I feel that this is not an appropriate consideration.
A). I made some revisions as follows according to your suggestion. It might be better than before.
In page 3 of 19
A sub-analysis showed that there was no difference of mean duration of antibiotic therapy between survival and 30-day death group among HCAP patients (survival 14.2 vs. 30-day death 10.3 days, p=0.21). The result of our study also suggests that HCAP patients are likely to have longer duration of antibiotic therapy 10-14 days as we expected. The therapeutic strategy for HCAP patients might have to be reconsidered due to the poor general conditions.
- The Conclusion should also be reevaluated, I believe. Is it the only important point of this study that only 14% of pneumonia patients are eligible to participate in clinical trials? Isn't the important point that many patients with poor physical status, much complications or severe illness are not considered in the clinical trials?
A). I totally agree wit your opinion. I made some revisions as follows.
- Discussion
Patients in the real world are quite different from those who can participate in a clinical trial. We already reported that only 24% of candidemia patients could participate in a clinical trial. Patients who can participate in a clinical trial have better PSs and longer overall survival times than those seen in actual medical practice [2]. Like the study, community-onset pneumonia patients with participation possible group showed lesser severity of pneumonia and fewer comorbidities than those with the participation impossible group. We found the participation possible group had higher mortality rates of 30-day and in-hospital than the participation impossible group. However, identification of PDR pathogens, mechanical ventilation and Do Not Attempt Resuscitation (DNAR) order were more frequently seen in the participation possible group than in the impossible group. In Japan, discussing DNAR order with Japanese family members is still considered to be taboo [25]. Therefore, these results could suggest that patients in the participation impossible group have a worse prognosis than those in the participation possible group do. It is well known that HCAP patients are more likely to have worse PSs and more comorbidities than those with CAP [4,5,7]. We should consider a RCT focusing on the elderly or fragile people who are usually excluded from the trials or analyze alternatives such as propensity score matching analysis. These will be helpful for clinicians to make a rational decision in treating those patients.

Round 2
Reviewer 1 Report
Thank you for the corrections made to the manuscript.
However, not all points have been taken into account :
1-
"Tables :
Please specify "other Enterobacteriaceae" rather than Enterobacteriaceae, as E coli, K; pneumoniae, etc.. are also Enterobacteriaceae.
there are still many typing errors in the names of bacteria (text and tables), please check meticulously in the whole manuscript."
Thank you to the authors to proofread meticulously and rigourously in the whole manuscript, if necessary with the help of a microbiologist.
In addition, some pathogens added under the term "Enterobacteriacea" in the revised table annotation are not Enterobacteriaceae.
2- Please add the management of missing values in the method section, in order to make this study reproducible.
Author Response
Manuscript ID: jcm-1483922      Date: Jan 1, 2022
Title: A large gap of patients' characteristics and outcomes between the real-world and clinical trial settings in community-acquired pneumonia and healthcare-associated pneumonia
Thank you very much for your great reviewing. We totally agree with your opinions and made appropriate revisions according to your suggestions. The revised portion was red in the text. We strongly believe that this revision made the text better than before. Please, consider my article for publication in your journal.
Sincerely,
Nobuhiro Asai and Hiroshige Mikamo
Reviewer 1
Comments and Suggestions for Authors
Thank you for the corrections made to the manuscript.
However, not all points have been taken into account :
1-
"Tables :
Please specify "other Enterobacteriaceae" rather than Enterobacteriaceae, as E coli, K; pneumoniae, etc.. are also Enterobacteriaceae.
there are still many typing errors in the names of bacteria (text and tables), please check meticulously in the whole manuscript."
Thank you to the authors to proofread meticulously and rigourously in the whole manuscript, if necessary with the help of a microbiologist.
In addition, some pathogens added under the term "Enterobacteriacea" in the revised table annotation are not Enterobacteriaceae.
A). I totally agree with your suggestions. E. coli and K. pneumonia are Enterobacteriaceae. Then, I revised the table and figures.
Enterobacteriaceae is replaced to “Other Enterobacteriaceae”.
2- Please add the management of missing values in the method section, in order to make this study reproducible.
A). Thank you for your suggestion. I agree with it and then added the following sentence.
In Method section
2.5. Statistical analyses
The data for categorical variables are expressed as percentages and continuous variables as mean ± standard deviation (SD). Chi-square or Fisher’s exact test (two-tailed) was used to compare categorical variables and unpaired Student’s t test or Mann–Whitney U test to compare continuous variables. Overall survival time (OS) was calculated as from the date of diagnosis until the date of death from any cause. A significance was evaluated by Log-Rank-test. Missing values were evaluated by the missing analysis of the software. Statistical analyses involved use of SPSS version 26 for Windows (SPSS Inc., Chicago, IL, USA). A p-value <0.05 was considered statistically significant.

Reviewer 2 Report
I have no further comments
Author Response
Manuscript ID: jcm-1483922      Date: Jan 01, 2022
Title: A large gap of patients' characteristics and outcomes between the real-world and clinical trial settings in community-acquired pneumonia and healthcare-associated pneumonia
Thank you very much for your great reviewing. We totally agree with your opinions and made appropriate revisions according to your suggestions. The revised portion was red in the text. We strongly believe that this revision made the text better than before. Please, consider my article for publication in your journal.
Sincerely,
Nobuhiro Asai and Hiroshige Mikamo
Reviewer 2
Comments and Suggestions for Authors
I have no further comments
A). I really appreciate your great reviewing and feedback. Thank you so much.
NOBUHIRO ASAI

Reviewer 3 Report
I feel that the authors have responded appropriately to my previous peer review report. The quality of the manuscript has improved, including responses to other reviewers' suggestions, and I thought it was worthy for publication.
Author Response
Manuscript ID: jcm-1483922      Date: Jan 01, 2022
Title: A large gap of patients' characteristics and outcomes between the real-world and clinical trial settings in community-acquired pneumonia and healthcare-associated pneumonia
Thank you very much for your great reviewing. We totally agree with your opinions and made appropriate revisions according to your suggestions. The revised portion was red in the text. We strongly believe that this revision made the text better than before. Please, consider my article for publication in your journal.
Sincerely,
Nobuhiro Asai and Hiroshige Mikamo
Reviewer 3
Comments and Suggestions for Authors
I feel that the authors have responded appropriately to my previous peer review report. The quality of the manuscript has improved, including responses to other reviewers' suggestions, and I thought it was worthy for publication.
A). I really appreciate your great reviewing and feedback. I believe that the revision according to your suggestion made the article much better than before. Thank you so much.
NOBUHIRO ASAI
